# Evolutionary and functional analyses reveal a role for the RHIM in tuning RIPK3 activity across vertebrates

Elizabeth J Fay, Kolya Isterabadi, Charles M Rezanka, Jessica Le, Matthew D Daugherty*

Department of Molecular Biology, School of Biological Sciences, University of California, San Diego, La Jolla, United States

## eLife Assessment

This **important** study provides **compelling** evidence for the evolutionary diversification and conserved NFκB-inducing function of RHIM-containing RIP kinase proteins across animal lineages, combining thorough bioinformatic analysis with functional assays in human cells. The findings are of broad interest to immunologists and evolutionary biologists, though some novel observations would benefit from deeper conceptual integration.

*For correspondence:
mddaugherty@ucsd.edu

Competing interest: The authors declare that no competing interests exist.

**Abstract** Receptor interacting protein kinases (RIPK) RIPK1 and RIPK3 play important roles in diverse innate immune pathways. Despite this, some RIPK1/3-associated proteins are absent in specific vertebrate lineages, suggesting that some RIPK1/3 functions are conserved, while others are more evolutionarily labile. Here, we perform comparative evolutionary analyses of RIPK1–5 and associated proteins in vertebrates to identify lineage-specific rapid evolution of RIPK3 and RIPK1 and recurrent loss of RIPK3-associated proteins. Despite this, diverse vertebrate RIPK3 proteins are able to activate NF-κB and cell death in human cells. Additional analyses revealed a striking conservation of the RIP homotypic interaction motif (RHIM) in RIPK3, as well as other human RHIM-containing proteins. Interestingly, diversity in the RIPK3 RHIM can tune activation of NF-κB while retaining the ability to activate cell death. Altogether, these data suggest that NF-κB activation is a core, conserved function of RIPK3, and the RHIM can tailor RIPK3 function to specific needs within and between species.

## Introduction

Receptor interacting protein kinase 3 (RIPK3) and the closely related kinase RIPK1 have critical roles in mediating cell death and inflammatory signaling (**Newton, 2015**). Downstream of signals from innate immune receptors like Z-DNA binding protein 1 (ZBP1) and tumor necrosis factor receptor 1 (TNFR1), RIPK3, and RIPK1 interact via their shared RIP homotypic interaction motif (RHIM) to form amyloid-like aggregates (**Wu et al., 2021**), leading to their autophosphorylation and activation of additional effector proteins to carry out their innate immune functions. For example, RIPK3 can phosphorylate MLKL to activate highly inflammatory programmed necrotic cell death (necroptosis) (**Sun et al., 2012**), RIPK1/3 can engage caspase-8 (CASP8) to activate apoptotic caspases (**Wang et al., 2008**; **Mandal et al., 2014**), or RIPK1/3 can activate the pro-survival and pro-inflammatory transcription factor NF-κB (**Meylan et al., 2004**; **Moriwaki et al., 2014**). Several factors are known to impact the outcome of RIPK1/3 activation, such as the activating signal and caspase activity (**Newton, 2015**;

*Orozco and Oberst, 2017*). Altogether, RIPK3 and RIPK1 are central to determining cell fate downstream of various innate immune stimuli.

Like many innate immune proteins, RIPK3 is evolutionarily dynamic in vertebrates. Phylogenomic analyses of a small number of species from different vertebrate clades suggest that RIPK3 and the associated proteins ZBP1 and MLKL have been lost in some vertebrate lineages, including birds, carnivores, and marsupials (*Dondelinger et al., 2016*; *Águeda-Pinto et al., 2021*). In addition, RIPK3, RIPK1, MLKL, and ZBP1 are known to interact with viral proteins (*Harris et al., 2015*; *Croft et al., 2018*; *Wagner et al., 2015*; *Muscolino et al., 2021*; *Huang et al., 2015*; *Liu et al., 2018*) and have been shown to be evolving under recurrent positive selection in both primates and bats (*Palmer et al., 2021*; *Cariou et al., 2022*), a hallmark of host proteins that directly interact with viral proteins (*Daugherty and Malik, 2012*). Despite the known evolutionary divergence of RIPK3 in vertebrates, most of what is known about their activation and regulation is from studies conducted in humans and mice. Even between humans and mice, there are known differences in the roles of specific genes in the regulation of NF-κB, suggesting this pathway can be tailored to the distinct environments of different species (*Zhang et al., 2017*). These differences likely extend beyond humans and mice and likely include RIPK1/RIPK3-mediated activation of NF-κB. Furthermore, RIPK1 and RIPK3 are part of a larger family of RIPKs, including RIPK2, RIPK4, and ANKK1/RIPK5, which, with the exception of RIPK5, are known to activate NF-κB in various cellular contexts (*Cuny and Degterev, 2021*; *Eng et al., 2021*; *Lv et al., 2022*). The degree to which these other RIP kinases and their interacting partners have diversified across vertebrates is unknown.

Here, we use evolutionary and functional approaches to characterize the diversification of RIPK3-mediated activation of NF-κB. We identified RIPK1- and RHIM-mediated activation of NF-κB by diverse vertebrate RIPK3 proteins, as well as distinct mechanisms of RIPK3-mediated activation of NF-κB. Phylogenetic and phylogenomic analyses of RIPK1–5 and several key associated proteins revealed the dynamic evolution of RIPK3 and RIPK1. In addition to observing the loss of RIPK3 and necroptosis-associated proteins in some vertebrate lineages, we identify changes in regulatory features—including catalytic sites and CASP cleavage sites—in RIPK3 and RIPK1 in specific mammalian lineages that may affect their function. Intriguingly, we found that the RHIM domain of RIPK3 and RIPK1, as well as other RHIM-containing proteins, is highly conserved in vertebrates and in a non-vertebrate RIPK1 protein. Consistent with this strong conservation of the RHIM domain, we found that RIPK1 proteins from diverse vertebrates, and even a non-vertebrate RIPK1, are able to activate NF-κB, suggesting that activation of this inflammatory response is a conserved function of RIPK3, RIPK1, and potentially other RHIM-containing proteins in vertebrates. Altogether, these data suggest that activation of NF-κB is the ancestral and conserved feature of RIPK3 and RIPK1, while other functions such as necroptosis are more evolutionarily labile and rapidly evolving, likely as a response to evolutionary pressure from pathogens.

## Results

### Comparative evolutionary analysis of RIPK1–5

To understand the evolution and diversification of RIPK function, we first wished to compare the functions of human RIPKs. There are canonically four RIPKs in humans, RIPK1–4. ANKK1 is a close paralog of RIPK4 and has therefore been denoted as RIPK5. Additional proteins have also been described as RIPK proteins, including DSTYK, LRRK1, and LRRK2, but these proteins are phylogenetically distinct and will therefore not be considered here (*Lv et al., 2022*). The shared domain architecture of RIPK1–5 includes a conserved N-terminal kinase domain, an extended disordered intermediate domain, and distinct protein interaction domains at the C-terminus that can include a death domain (DD), RHIM, caspase activation and recruitment domain (CARD), or ankyrin repeats (*Figure 1A*). Human RIPK1–4 activate NF-κB, albeit through different mechanisms, including differential dependence on endogenous RIPK1 and kinase activity (*Figure 1—figure supplements 1–3*). Interestingly, despite its close homology to RIPK4, RIPK5 did not activate NF-κB. Altogether, these data highlight NF-κB as a shared function of RIPK1–4.

We next wished to perform comparative evolutionary analyses of RIPK1–5 in vertebrates to determine which RIPKs might be undergoing functional divergence. We first analyzed RIPK1–5 in four distinct mammalian clades to determine whether there was evidence of recurrent positive selection,

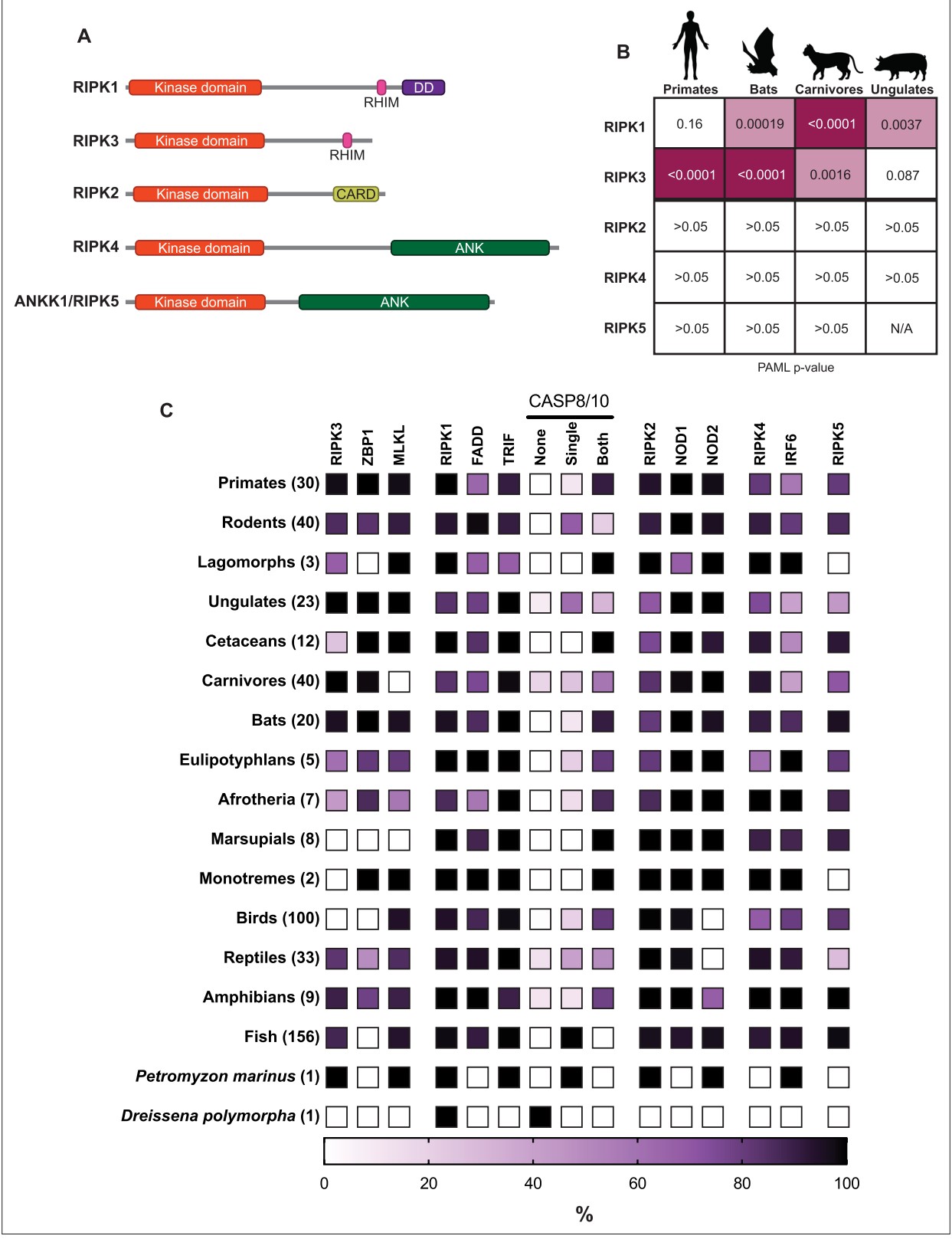

**Figure 1.** Comparative evolutionary analysis of RIPK1–5. (**A**) Domain structures of human RIP kinases. RHIM = RIP homotypic interaction motif, DD = death domain, CARD = caspase activation and recruitment domain, ANK = ankyrin repeats. (**B**) Positive selection analysis of RIPK1–5 in the indicated mammalian order. Input sequences and PAML p-values can be found in *Supplementary files 1 and 2*. Images of model species were generated using BioRender.com. (**C**) Heat map showing the percentage of species within a clade that have the indicated protein. The clades and the number of species

*Figure 1 continued on next page*

*Figure 1 continued*

within each clade are indicated on the left. Complete lists of proteins and species in each group can be found in **Supplementary files 4 and 5**. Lancelet species were not queried for all proteins and are not included in this graph.

The online version of this article includes the following source data and figure supplement(s) for figure 1:

**Figure supplement 1.** Human RIPK1–4 activate NF-κB.

**Figure supplement 1—source data 1.** Raw data for the bar graphs in **Figure 1—figure supplement 1** depicting NF-κB activation by various human RIPK proteins.

**Figure supplement 2.** Generation of RIPK1 KO HEK293T cells.

**Figure supplement 2—source data 1.** Original western blots for **Figure 1—figure supplement 2**.

**Figure supplement 2—source data 2.** PDF file containing original western blots for **Figure 1—figure supplement 2**, indicating the relevant bands and treatments.

**Figure supplement 3.** Expression of RIPK1–5 constructs.

**Figure supplement 3—source data 1.** Original western blots for **Figure 1—figure supplement 3**.

**Figure supplement 3—source data 2.** PDF file containing original western blots for **Figure 1—figure supplement 3**, indicating the relevant bands and treatments.

**Figure supplement 4.** Phylogenomic analysis of RIPK and associated proteins.

---

which can be a hallmark of host-pathogen interactions and can drive functional diversification. Notably, we only found evidence for positive selection acting on RIPK1 and RIPK3 (**Figure 1B**, **Supplementary files 1 and 2**). Interestingly, RIPK1 and RIPK3 are not uniformly evolving under positive selection in all mammalian clades analyzed. For instance, in primates, RIPK3 is evolving under strong positive selection, while RIPK1 is not, consistent with previous data (**Palmer et al., 2021**; **Cariou et al., 2022**). In contrast, ungulate RIPK3 does not show evidence for positive selection, whereas ungulate RIPK1 does. These data indicate that among RIPK1–5, only RIPK1 and RIPK3 genes have undergone recurrent positive selection, and that RIPK1 and RIPK3 may have faced different selective pressures across distinct mammalian lineages.

In addition to evolving under positive selection in primates and bats (**Palmer et al., 2021**), RIPK3, ZBP1, and MLKL are known to be lost in some vertebrate lineages (**Dondelinger et al., 2016**). However, it is unknown if other RIPK and RIPK-associated proteins are similarly lost. We therefore performed phylogenomic analysis on 489 vertebrate species for the presence or absence of RIPK1–5 and several key RIPK-associated proteins (**Figure 1C**, **Supplementary files 4 and 5**). Due to the difficulty in discerning whether a gene absence in an individual species is due to true gene loss or incomplete genome assembly, we focused on multispecies patterns of gene loss within and between vertebrate lineages. Given that NF-κB is broadly present in many vertebrate and non-vertebrate species, including horseshoe crab (**Wang et al., 2006**), *Drosophila melanogaster* (**Hetru and Hoffmann, 2009**), and cnidarians and bivalves (reviewed in **Williams and Gilmore, 2020**), we analyzed proteins associated with other RIPK functions. We first confirmed the loss of necroptosis-associated proteins that has been previously reported (**Dondelinger et al., 2016**): RIPK3 and ZBP1 in birds; RIPK3, ZBP1, and MLKL in marsupials; MLKL in carnivores. In addition, we found that ZBP1 is absent in all fish species, suggesting that this protein, and ZBP1-mediated necroptosis, arose only during tetrapod evolution.

We then analyzed RIPK1 and its associated proteins. RIPK1 is found nearly every vertebrate genome queried, and clear RIPK1 homologs were identified in several non-vertebrate species, including zebra mussel (*Dreissena polymorpha*), several additional spiralia species, and lancelet species in the genus *Branchiostoma* (accession numbers in **Supplementary file 4**). RIPK1 homologs—defined as proteins with a homologous N-terminal kinase domain and a C-terminal RHIM tetrad and DD—were not found in tunicates, echinoderms, cnidarians, nematodes, arthropods, or amoeba. Additional RIPK-like proteins have been found in other non-vertebrate species, including a RIPK1/2-like protein in *Drosophila* and other protostomes (**Dondelinger et al., 2016**), although these proteins were not considered here due to minimal sequence similarity to the human RIP kinase domains and our focus on vertebrate RIP kinases. We also identified CASP8, a known regulator of RIPK1, and its closest homolog, CASP10. Due to the similar domain structure of CASP8 and CASP10, we only identified whether a species has a single CASP8/10 protein, two distinct

CASP8/10 proteins as determined by phylogeny (see Materials and methods), or none. At least one copy of CASP8/10 is found in most vertebrate genomes. Two copies, presumably marking the delineation between CASP8 and CASP10, began in tetrapod, coinciding with the emergence of ZBP1. This may reflect the role of CASP8 in regulating ZBP1, and duplication of this caspase allowed for the specialized function of CASP8 (*Rodriguez et al., 2022*). We also confirmed the loss of CASP10 in some rodents (*Eckhart et al., 2008*) (see *Figure 1—figure supplement 4A* for CASP8/10 breakdown), revealing that suborder Myomorpha ('mouse-like') and Castorimorpha ('beaver-like') rodents have lost CASP10, whereas Sciuromorpha ('squirrel-like') and Hystricomorpha ('porcupine-like') rodents have retained CASP10. Finally, we found that FADD and TRIF are well conserved in all vertebrates.

We also analyzed the phylogenomic distribution of RIPK2, -4, and -5 and their associated proteins. RIPK2 and its associated protein NOD1 are found in most vertebrates (*Cuny and Degterev, 2021*; *Eng et al., 2021*). Intriguingly, we observed a second paralog of RIPK2, which we call RIPK2B, in many vertebrates but lost in therian (live-bearing) mammals (*Figure 1—figure supplement 4B*). NOD2 is present in lamprey and fish, but absent in birds, reptiles, and some amphibians, including some of the same species that contain RIPK2B. While RIPK2B retains both the kinase domain and CARD, there may be distinct regulatory mechanisms, specifically those related to NOD2, compared to RIPK2. RIPK4 is present in jawed vertebrates. IRF6, the only known phosphorylation target of RIPK4 (*Cuny and Degterev, 2021*; *Eng et al., 2021*), emerged prior to RIPK4 and has been lost in many vertebrate species, particularly in carnivores, cetaceans, and ungulates. RIPK5, which has no known function and no known interacting proteins, has been lost in several lineages, including monotremes and lagomorphs. Overall, our analyses revealed that RIPK1 and RIPK3 are evolutionarily distinct from other RIP kinases, and the recurrent loss of some RIPK1/3-associated necroptosis proteins (ZBP1, MLKL, CASP8) and RIPK3 suggests that these proteins may have functions tailored to the needs of specific vertebrate lineages.

## Diverse vertebrate RIPK3 proteins activate NF-κB

Our analyses revealed dynamic evolution of RIPK3 in vertebrates, including signatures of positive selection in multiple mammalian clades and recurrent loss of RIPK3 and its associated proteins. We hypothesized that this rapid RIPK3 evolution may have resulted in different functions of RIPK3 in different vertebrate lineages. We therefore tested the ability of RIPK3 proteins from diverse vertebrates, ranging from 23% to 63% sequence identity to humans (*Figure 2A*, *Figure 2—figure supplement 1*), to activate NF-κB in human cells. We included two mammalian species that retain all necroptosis proteins (mouse, *Mus musculus*; pig, *Sus scrofa*), a mammalian species lacking a necroptosis protein (cat, *Felis catus*), reptilian species that do (turtle, *Chelonia mydas*) and do not (lizard, *Anolis carolinensis*) retain all necroptosis proteins (*Supplementary file 7*), and sea lamprey (*Petromyzon marinus*) as our most distal vertebrate relative to humans. To our surprise, among all vertebrates tested, only pig and sea lamprey RIPK3 failed to activate NF-κB in human cells (*Figure 2B*). This striking conservation of function indicates that NF-κB activation has remained intact despite lineage-specific RIPK3 divergence and loss of interacting partners. We then analyzed whether, like human RIPK3 (*Figure 1—figure supplement 1C*; *Yatim et al., 2015*), other vertebrate RIPK3 proteins require RIPK1 to activate NF-κB by repeating experiments in RIPK1 KO cells. Interestingly, while mammalian RIPK3 activation is dependent on RIPK1, both lizard and turtle RIPK3 activate NF-κB independently of RIPK1 (*Figure 2B*). These results indicate that the requirements for NF-κB activation by RIPK3 are species-specific.

Finally, we tested whether another function of RIPK3, activation of ZBP1- and MLKL-dependent cell death (*Figure 2—figure supplement 2*), is also conserved. Most mammalian RIPK3 proteins were able to activate cell death using human ZBP1 and MLKL (*Figure 2C*). This included pig RIPK3, which is unable to activate NF-κB, suggesting that activation of cell death and NF-κB by RIPK3 proteins has been separated in a species-specific manner. Likewise, we observed that cat RIPK3 can activate NF-κB but fails to activate cell death. Based on this observation, we analyzed the sequences of carnivore RIPK3 proteins and found that feline RIPK3 has mutated kinase catalytic residue (*Figure 2—figure supplement 3*), explaining the loss of cell death activation by cat RIPK3. Outside of mammals, only lizard RIPK3 was able to activate cell death, despite the ability of both lizard and turtle RIPK3 to activate NF-κB, further suggesting that these functions of RIPK3 have distinct, species-specific requirements.

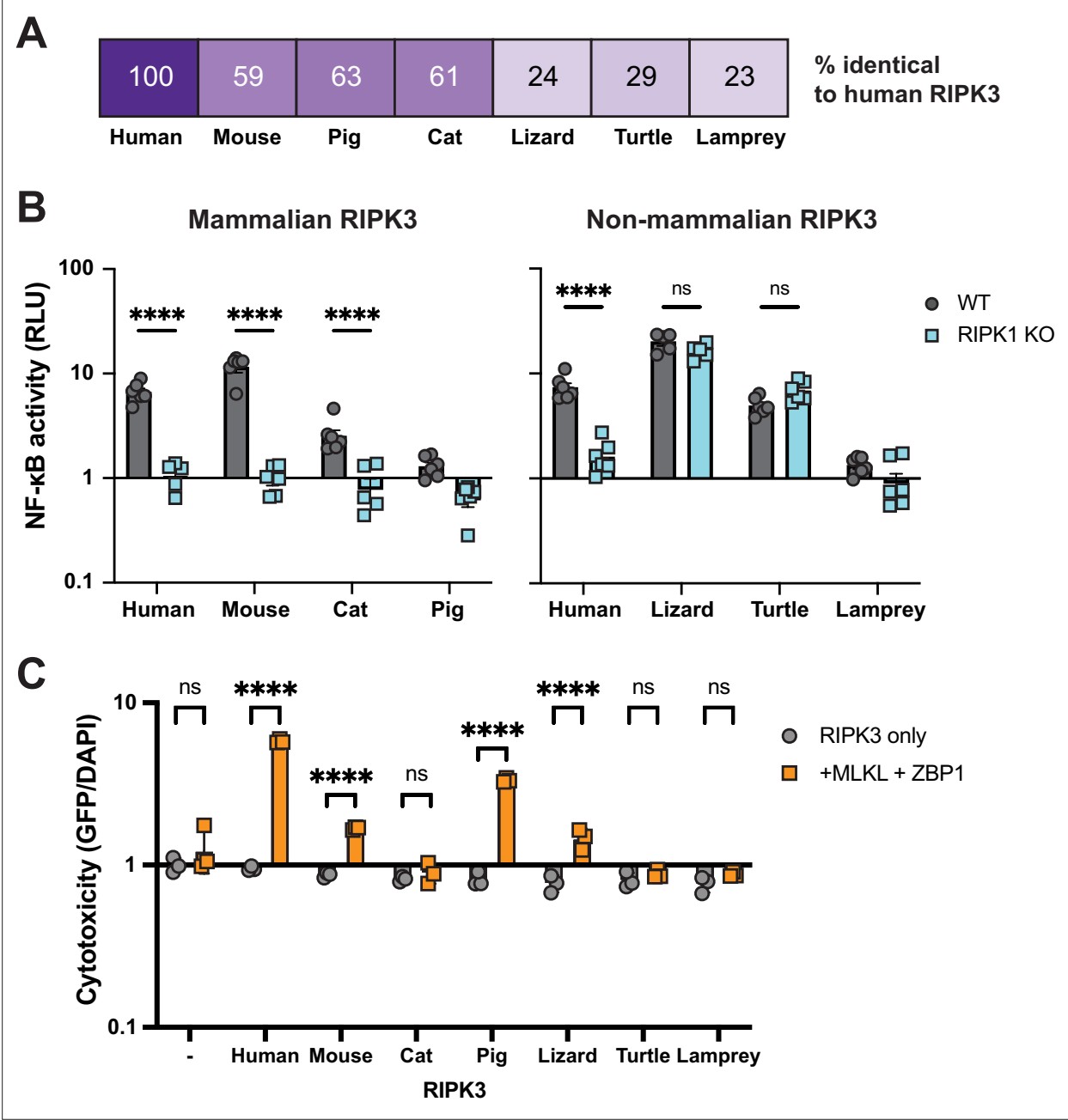

**Figure 2.** Diverse vertebrate RIPK3 proteins activate NF-κB. (**A**) Percent similarity of RIPK3 from the indicated species compared to humans. (**B**) RIPK3 proteins were transfected into WT or RIPK1 KO HEK293T cells, along with NF-κB firefly luciferase and control renilla luciferase reporter plasmids (see Materials and methods), and NF-κB activity was measured at 18 hr post-transfection. (**C**) RIPK3 proteins were transfected into HEK293T cells with and without human ZBP1 and MLKL. At 18 hr post-transfection, cells were stained using the ReadyProbe Cell Viability kit, and fluorescence was measured using a plate reader. Species shown are mouse (*M. musculus*), cat (*F. catus*), pig (*S. scrofa*), lizard (*A. carolinensis*), turtle (*C. mydas*), and lamprey (*P. marinus*). Data are representative of 3–5 independent experiments with n=3–6 replicates per group. Data were analyzed using two-way ANOVA with Šidák's multiple comparisons test. ns = not significant, ****=p<0.0001.

The online version of this article includes the following source data and figure supplement(s) for figure 2:

**Source data 1.** Raw data for the bar graphs in *Figure 2* depicting NF-κB (B) and cell death (C) activation by various vertebrate RIPK3 proteins.

**Figure supplement 1.** Expression of nonhuman RIPK3 proteins.

**Figure supplement 1—source data 1.** Original western blots for *Figure 2—figure supplement 1*.

**Figure supplement 1—source data 2.** PDF files containing original western blots for *Figure 2—figure supplement 1*, indicating the relevant bands and treatments.

*Figure 2 continued on next page*

*Figure 2 continued*

**Figure supplement 2.** MLKL- and ZBP1-dependent cell death activation by RIPK3.

**Figure supplement 2—source data 1.** Raw data for the bar graph in *Figure 2—figure supplement 2* depicting cell death activation by human MLKL, ZBP1, and/or RIPK3.

**Figure supplement 3.** Loss of RIPK3 catalytic site in carnivores.

## Conservation of the RHIM sequence determines RIPK3 activation of NF-κB

The conservation of NF-κB activation by RIPK3 proteins that share as little as 24% sequence identity and are up to >300 million years diverged (*Benton and Donoghue, 2007*) suggests some regions of the protein are likely highly conserved. To identify such regions, we returned to our positive selection analyses. While we identified rapidly evolving sites throughout the primate RIPK3, there was a cluster of sites in and around the C-terminal RHIM (*Figure 3*, *Supplementary file 5*), which is a known region of interaction between RIPK1 and RIPK3 and RIPK3 homooligomerization (*Wu et al., 2021*; *Sun et al., 2002*). Intriguingly, despite multiple rapidly evolving sites within the RHIM, a core tetrad (VQVG) is not rapidly evolving. Similarly, the core tetrads of bat and carnivore RIPK3 are not rapidly evolving (*Figure 3B*). Furthermore, the RIPK3 core RHIM tetrad is highly conserved across vertebrates (*Figure 3C*). This is despite the loss of other functional motifs in specific lineages, including the loss of the RIPK3 catalytic site in felines (*Figure 2—figure supplement 3*).

We therefore tested whether this conserved RHIM tetrad was responsible for activation of NF-kB by diverse vertebrate RIPK3s. We mutated the RIPK3 RHIM core tetrad (V/I-Q-V/I-G to AAAA) in our panel of vertebrate RIPK3 proteins and tested the ability of these proteins to activate NF-kB in human cells. Mutation of human, mouse, and cat RIPK3 RHIMs reduced activation of NF-κB as expected (*Figure 3D*, *Figure 2—figure supplement 1*). Surprisingly, despite activating independently of RIPK1 (*Figure 2B*), activation of NF-κB by lizard and turtle RIPK3 remains dependent on the RHIM (*Figure 3E*, left) and independent of RIPK3 kinase activity (*Figure 3—figure supplements 1 and 2*). These data support that reptile RIPK3 activates NF-κB through a distinct mechanism from mammalian RIPK3, but one that is still highly dependent on the RHIM core tetrad. Interestingly, while analyzing RHIM domains of nonhuman RIPK3, we observed that the sea lamprey RHIM diverges from the canonical V/I-Q-V/I-G tetrad and instead contains TQIG (*Figure 3C*). To determine if this was responsible for the limited activation of NF-κB by sea lamprey RIPK3, we mutated this motif to IQIG and observed an increase in NF-κB activation (*Figure 3E*, right). Altogether, these data indicate that NF-κB activation is a conserved function of RIPK3 that requires conservation of the core RHIM tetrad but allows for substantial sequence divergence elsewhere in the protein.

## NF-κB activation is a shared function of RHIM-containing proteins and can be tuned by the RHIM

Given that activation of NF-κB by RIPK3 is dependent on the RHIM, we hypothesized that the core RHIM tetrad and activation of NF-κB would be broadly conserved across all RHIM-containing proteins, including RIPK1, ZBP1, and TRIF. We found that the RIPK1 RHIM is highly conserved in vertebrates and in RIPK1-like proteins found in zebra mussel, *Lingula* species, and multiple lancelet species (*Figure 4A*). Human RIPK1 is known to activate NF-κB independently of its RHIM through polyubiquitination of the intermediate domain (*Li et al., 2006*; *Ea et al., 2006*). However, the strong conservation of the RHIM domain tetrad sequence (V/I-Q-V/I-G) of RIPK1 across species led us to hypothesize that RHIM-mediated activation of NF-κB by RIPK1 is also conserved. We therefore tested a diverse panel of metazoan RIPK1 proteins in our NF-κB assay. To avoid the effects of both kinase activity and kinase-mediated interactions (*Delanghe et al., 2020*), we generated RIPK1 C-terminus (RIPK1$^{CT}$) proteins from diverse species that do (mouse, pig) and do not (cat, lizard, lamprey, zebra mussel) retain all necroptosis proteins (*Figure 4—figure supplement 1*). Both WT and RHIM mutant human RIPK1$^{CT}$ activate NF-κB to the same extent as full-length RIPK1 (*Figure 4—figure supplement 2*), validating our use of this system to characterize RIPK1 functions independent of the kinase domain. Diverse vertebrate RIPK1$^{CT}$ proteins activate NF-κB independent of endogenous RIPK1 (*Figure 4—figure supplement 3*), consistent with our hypothesis that NF-κB activation is an ancestral and conserved function of these proteins. However, like RIPK3, the specific sequence requirements for RIPK1-mediated NF-κB

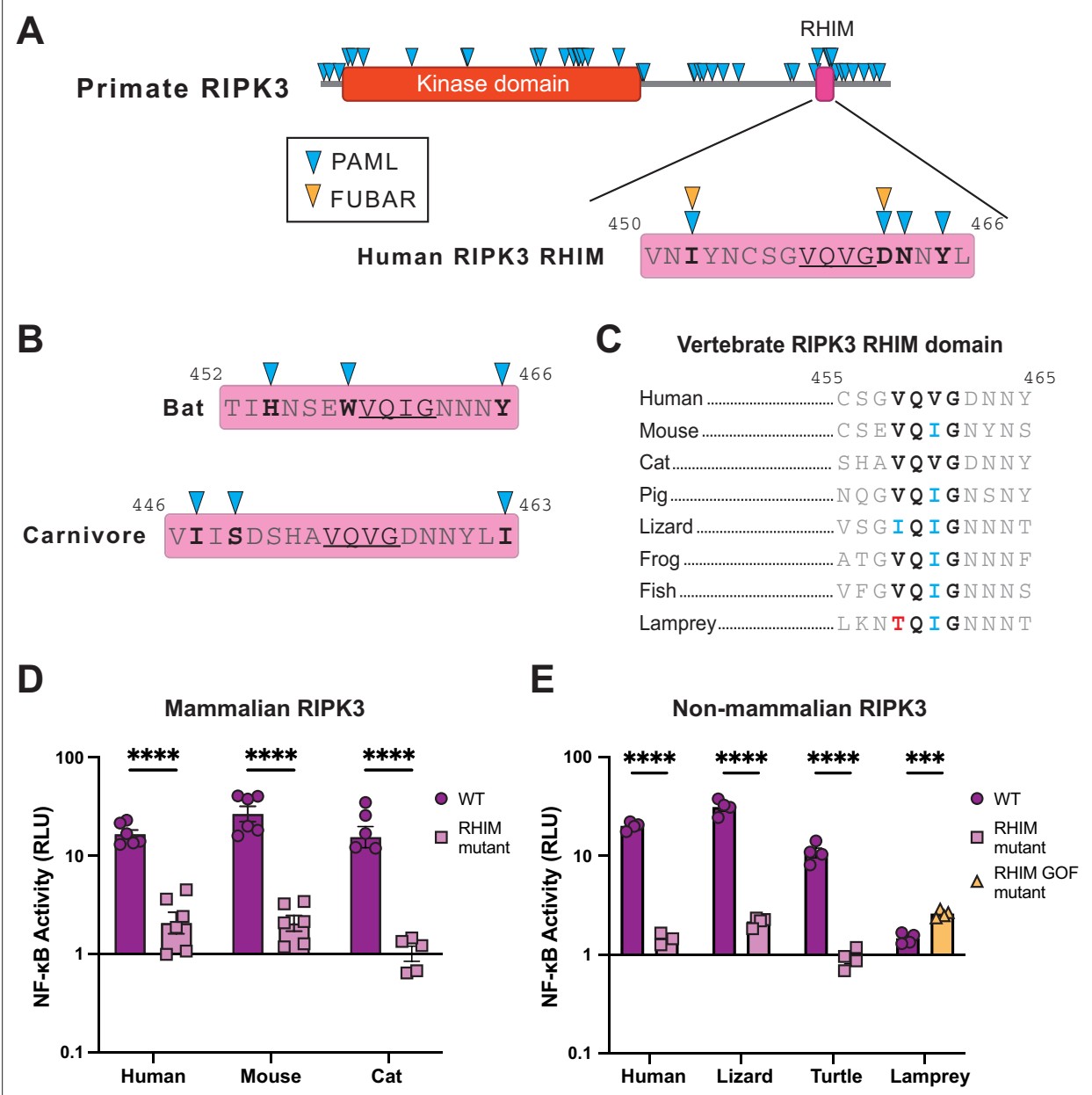

**Figure 3.** Conservation of the RHIM sequence determines RIPK3 NF-κB activation. (**A**) Residues evolving under positive selection as identified using PAML and FUBAR (see Materials and methods) in primate RIPK3 and the primate RIPK3 RHIM domain mapped on to the human sequence. (**B**) Residues evolving under positive selection in bat and carnivore RIPK3, mapped on to the *Sturnira hondurensis* and *F. catus* sequences, respectively. (**C**) Alignments of the RIPK3 RHIM across diverse vertebrates. Residue numbers refer to the human sequence. (**D–E**) Mammalian (**D**) and nonmammalian (**E**) RIPK3 proteins were transfected into WT HEK293T cells along with NF-κB firefly luciferase and control renilla luciferase reporter plasmids (see Materials and methods). NF-κB activity was measured at 18 hr post-transfection. Species shown are mouse (*M. musculus*), cat (*F. catus*), pig (*S. scrofa*), lizard (*A. carolinensis*), turtle (*C. mydas*), and lamprey (*P. marinus*). Data are representative of 3–5 independent experiments with n=3–6 replicates per group. Data were analyzed using two-way ANOVA with Šidák's multiple comparisons test. ns = not significant, ****=p<0.0001.

The online version of this article includes the following source data and figure supplement(s) for figure 3:

**Source data 1.** Raw data for the bar graphs in *Figure 3* depicting cell death activation by various wild-type and RHIM mutant vertebrate RIPK3 proteins.

**Figure supplement 1.** RIPK1-independent activation of reptile RIPK3 is independent of kinase activity.

**Figure supplement 1—source data 1.** Raw data for the bar graphs in *Figure 3—figure supplement 1* depicting cell death activation by wild-type, kinase mutant, and RIPK3 proteins.

**Figure supplement 2.** Expression of reptile RIPK3 kinase mutants.

*Figure 3 continued on next page*

*Figure 3 continued*

**Figure supplement 2—source data 1.** Original western blots for *Figure 3—figure supplement 2*.

**Figure supplement 2—source data 2.** PDF file containing original western blots for *Figure 3—figure supplement 2*, indicating the relevant bands and treatments.

activation varied by species. Specifically, mutation of the RHIM domain does not affect activation of NF-κB by mammalian proteins, including human, mouse, cat, and pig (*Figure 4B*, left). In contrast, mutation of the RHIM domain attenuated or completely prevented the ability of non-mammalian (lizard, sea lamprey, and zebra mussel) RIPK1$^{CT}$ to activate NF-κB (*Figure 4B*, right).

Other than RIPK1 and RIPK3, only two other human proteins contain a RHIM domain, TRIF and ZBP1. The TRIF RHIM and the first RHIM of ZBP1 are also highly conserved in mammals and some other tetrapods, whereas the second RHIM of ZBP1 diverges even within mammals (*Figure 4C*, *Figure 4—figure supplement 4A*). We then tested for RHIM dependence of NF-κB activation by human TRIF and ZBP1. As described previously (*Rebsamen et al., 2009*), ZBP1 activates NF-κB in a RHIM- and RIPK1-dependent manner (*Figure 4D*). Consistent with our evolutionary prediction, only the first RHIM tetrad is required for NF-κB activation by ZBP1 (*Figure 4D*). Conversely, mutation of the TRIF RHIM domain does not affect activation of NF-κB, and activation is only moderately reduced in RIPK1 KO cells (*Figure 4—figure supplement 4B and C*), likely due to the TIR domain in TRIF that is known to mediate its innate immune signaling (*Nanson et al., 2019*).

Finally, due to the central role of the core tetrad of the RHIM domain, we tested the plasticity of RHIM function with regard to sequence divergence. In our phylogenetic analyses of RIPK3, we identified several species in which the RHIM core tetrad has diverged, particularly outside of tetrapods, including tetrads that have diverged from the conserved motif (V/I-Q-V/I-G) at only a single residue (*Supplementary file 8*). To test for functional differences, we inserted these naturally occurring RHIM tetrads into human RIPK3 and characterized the ability to activate NF-κB and cell death. We included tetrads found in rodents, bats, eulipotyphlans, reptiles, and fish (VQFG), amphibians and fish (LQIG, VQSG), fish (CQIG), and sea lamprey (TQIG). Interestingly, while most tested variants activate NF-κB compared to the inactive RIPK3 RHIM mutant (AAAA tetrad), albeit lower than the WT (VQVG) tetrad, TQIG and VQSG did not activate NF-κB (*Figure 4E*, *Figure 4—figure supplements 5 and 6*). These data suggest that, while there is some plasticity in the RHIM tetrad, not all residues are functional. All tetrad variants were able to activate ZBP1-dependent cell death similar to WT RIPK3 (*Figure 4F*), further revealing the separation of RIPK3-mediated NF-κB activation from cell death activation. These differences may be due to differential amyloid structures formed during heterooligomerization of the RIPK1 RHIM and RIPK3 RHIM versus homooligomerization of RIPK3 (*Wu et al., 2021*). Altogether, these data suggest that diversity in the RIPK3 RHIM domain may tune activation of NF-κB and tailor RIPK3 function to the specific needs of species.

## Discussion

Activation of NF-κB is critical to both the innate and adaptive immune response in humans and mice (*Zhang et al., 2017*) and is a deeply conserved pathway in many metazoans (*Wang et al., 2006*; *Hetru and Hoffmann, 2009*; *Williams and Gilmore, 2020*). Despite this deep conservation of NF-κB signaling, differences in the function and regulation of NF-κB-associated proteins between mice and humans have been identified (*Zhang et al., 2017*), suggesting that this functionally well-conserved pathway can be adapted to different species. These data indicate that broader phylogenetic analysis of the genes and functions that are associated with NF-κB signaling is needed to understand immune responses across species, including those that harbor pathogens that pose a zoonotic threat to humans. Importantly, the approach used here can be applied to other NF-κB-associated proteins and to other innate immune pathways to better define species-specific immune responses.

Here, we apply broad phylogenetic and functional sampling to the RIP kinase family of NF-κB activators. Using a phylogenomic approach, we identified widespread conservation of RIPK1, RIPK2, and RIPK4 and their associated proteins within vertebrates. We also show conservation of activation of NF-κB signaling by RIPK1 and RIPK3 across diverse vertebrate species that is largely reliant on the highly conserved core tetrad of the RHIM domains found in both proteins. This striking conservation of RIPK1- and RIPK3-mediated activation of NF-κB, from species that span >500 million years of

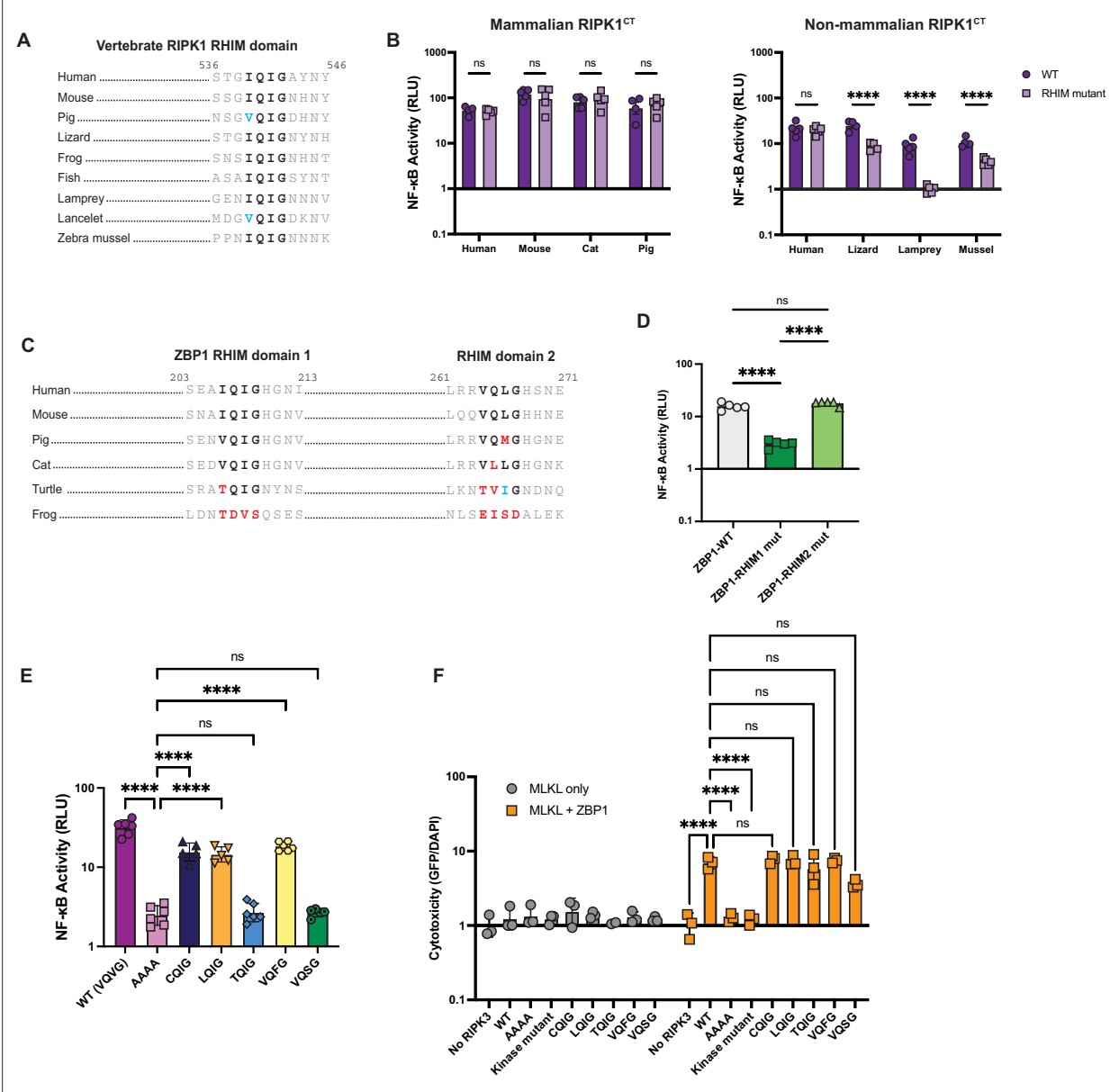

**Figure 4.** NF-κB activation is a shared function of RHIM-containing proteins and can be tuned by the RHIM. (**A**) Alignment of RIPK1 RHIM across diverse vertebrates. Residue numbers refer to the human sequence. Lancelet species is *Branchiostoma floridae*. (**B**) Diverse vertebrate RIPK1^CT proteins were transfected into HEK293T cells along with NF-κB firefly luciferase and control renilla luciferase reporter plasmids (see Materials and methods), and NF-κB activity was measured at 18 hr post-transfection. (**C**) Alignment of ZBP1 RHIMs across diverse vertebrates. Residue numbers refer to the human sequence. (**D**) Activation of NF-κB by WT and RHIM mutant ZBP1 proteins. (**E**) NF-κB activation by human RIPK3 with the indicated RHIM tetrad variant. (**F**) Human RIPK3 proteins with the indicated RHIM tetrad variants were transfected into HEK293T cells with MLKL (gray circles) or MLKL and ZBP1 (orange squares), and viability was measured at 18 hr post-transfection. Data are representative of 2–5 independent experiments with n=3–6 replicates per group. Data were analyzed using two-way ANOVA with Šidák's multiple comparisons test (**A, D**), one-way ANOVA with Tukey's multiple comparisons test (**E**), or two-way ANOVA with Tukey's multiple comparisons test (**F**). ns = not significant, ****=p<0.0001.

The online version of this article includes the following source data and figure supplement(s) for figure 4:

**Source data 1.** Raw data for the bar graphs in *Figure 4* depicting NF-κB (B, D, E) and cell death (F) activation by various RHIM-containing proteins.

**Figure supplement 1.** Expression of nonhuman RIPK1^CT proteins.

**Figure supplement 1—source data 1.** Original western blots for *Figure 4—figure supplement 1*.

**Figure supplement 1—source data 2.** PDF files containing original western blots for *Figure 4—figure supplement 1*, indicating the relevant bands and treatments.

*Figure 4 continued on next page*

*Figure 4 continued*

**Figure supplement 2.** RIPK1$^{CT}$ activates NF-κB similar to full-length RIPK1.

**Figure supplement 2—source data 1.** Raw data for the bar graphs in *Figure 4—figure supplement 2* depicting NF-κB activation by human RIPK1 proteins.

**Figure supplement 3.** Activation of NF-κB by diverse vertebrate RIPK1$^{CT}$ is independent of endogenous human RIPK1.

**Figure supplement 3—source data 1.** Raw data for the bar graphs in *Figure 4—figure supplement 3* depicting NF-κB activation by various vertebrate RIPK1$^{CT}$ proteins in wild-type and RIPK1 KO HEK293T cells.

**Figure supplement 4.** Activation of NF-κB by TRIF.

**Figure supplement 4—source data 1.** Raw data for the bar graphs in *Figure 4—figure supplement 4* depicting NF-κB activation by human TRIF proteins.

**Figure supplement 5.** Activation of NF-κB by human RIPK3 RHIM variants.

**Figure supplement 5—source data 1.** Raw data for the bar graphs in *Figure 4—figure supplement 5* depicting NF-κB activation by human RIPK3 proteins.

**Figure supplement 6—source data 1.** Original western blots for *Figure 4—figure supplement 6*.

**Figure supplement 6—source data 2.** PDF file containing original western blots for *Figure 4—figure supplement 6*, indicating the relevant bands and treatments.

**Figure supplement 6.** Expression of human RIPK3 RHIM variants.

---

vertebrate evolution, underscores the core functionality of RIP kinases as NF-κB activating proteins. Despite this conservation of NF-κB activation, we also observed lineage-specific changes in RIPK presence or mechanisms of NF-κB activation. For instance, we observed recurrent gene loss of RIPK3 and RIPK5 throughout the vertebrate phylogeny and discovered a second paralog of RIPK2 in many non-mammalian vertebrate species (RIPK2B). In addition, although RIPK3-mediated activation of NF-κB is highly dependent on RIPK1 across diverse mammals, RIPK3 activation of NF-κB in reptiles is independent of RIPK1. Activation of NF-κB is part of the antiviral immune response and is known to be antagonized by viruses across a range of species, including vertebrates and non-vertebrates (*Albarnaz et al., 2022*; *Gonzalez Lopez Ledesma et al., 2023*; *Palmer et al., 2019*). This antagonism could result in the adaptation of NF-κB-associated proteins, including RIP kinases, leading to lineage-specific mechanisms of activation that are tailored to the specific contexts within and between species.

The most striking differences in RIPK-associated genes and functions are those involved in necroptosis. Necroptosis is highly inflammatory, and the associated proteins must therefore be tightly regulated, as it could be a cause of immunopathology during infection. Despite this, necroptosis has been shown to be important for the response to several viruses (*Koehler et al., 2021*; *Balachandran and Rall, 2020*; *Shubina et al., 2020*). It is hypothesized that necroptosis, defined by RIPK3-mediated activation of MLKL, arose in vertebrates (*Dondelinger et al., 2016*; *Tummers and Green, 2022*). Interestingly, we find that two critical regulators of necroptosis, CASP8 and ZBP1, are only found in tetrapods, suggestive of an additional innovation within the necroptosis pathway during the divergence of tetrapods. Moreover, consistent with previous observations (*Dondelinger et al., 2016*; *Águeda-Pinto et al., 2021*), we observe a large number of necroptosis-associated gene losses, while also discovering important functional motifs in necroptosis proteins in distinct tetrapod lineages. These data support a model where necroptosis has been constructed as an important facet of the innate immune response in some tetrapods, but is not universal and has been deconstructed in species where the immunopathologic effects outweigh the benefits.

There have been several hypotheses to explain the loss of necroptosis-associated proteins in specific vertebrate lineages, such as diet (e.g. carnivores) or behavior (e.g. torpor in reptiles) (*Dondelinger et al., 2016*). While the cause of a specific instance of gene loss is difficult to identify, it is clear that gene loss can provide an evolutionary tool for adaptation (*Albalat and Cañestro, 2016*). For example, caspases, which are key regulators of both immunologically silent apoptosis and inflammatory pyroptosis, differ greatly across mammals, with lineage-specific caspase repertoires including both loss and gain of novel caspases (*Eckhart et al., 2008*). Even within a single mammalian order, homologous caspases have been found to serve different functions in response to pathogens (*Devant et al., 2021*). This type of lineage-specific innovation likely exists for necroptosis, and gene loss could be a cause or a result of adaptations. Indeed, differences in MLKL activation by RIPK3 across species have been identified (*Tanzer et al., 2016*), which may be the result of additional diverse regulatory mechanisms

for necroptosis even among species with an intact pathway. Strikingly, despite the lack of a complete necroptosis pathway in some lineages (cat, turtle, lamprey), our functional data indicate that RIPK3 retains the ability to activate NF-κB. Activation of NF-κB could therefore represent the ancestral function of RIPK3, and it has been co-opted to activate necroptosis in vertebrates.

The RHIM core tetrad was previously known to be conserved between humans and mice (*Riebeling et al., 2022*). Here, we expand this analysis and identify striking conservation of the RIPK3, RIPK1, ZBP1, and TRIF RHIM domains across most vertebrates. Additional RHIM-containing proteins have been identified in non-vertebrate animals and in fungi (*Kajava et al., 2014*). Interestingly, the RHIM proteins found in fruit fly and fungi are known to be involved in antibacterial defense and cell death, respectively, suggesting that RHIM-mediated interactions may be an ancient mechanism of innate immunity. While activation of NF-κB by all tested nonhuman RIPK3 proteins was highly dependent on the RHIM domain, only non-mammalian RIPK1$^{CT}$ proteins required the RHIM for maximal NF-κB activation. These species (lizard, lamprey, and zebra mussel) also lack ZBP1, with lamprey and zebra mussel predating the emergence of ZBP1 in tetrapods. Necroptosis may have driven the evolution of RHIM-independent activation of NF-κB by RIPK1. This may have also shaped RHIM-dependent functions of RIPK3 across vertebrates, as the RHIM tetrad is much less conserved in fish compared to tetrapods, and insertion of these tetrad variants into human RIPK3 alters its function. Virus antagonism has also likely played a role in the evolution of RIPK3 and RIPK1 function. For example, vaccinia virus E3 prevents Z-DNA sensing by ZBP1 (*Koehler et al., 2021*), human cytomegalovirus UL36 promotes MLKL degradation, and poxviruses encode an MLKL homolog to prevent RIPK3-mediated activation of host MLKL (*Petrie et al., 2019*). Conservation of the RHIM domain may be a way to retain some RIPK3/RIPK1 innate immune functions in the face of such antagonism. However, several viruses have taken advantage of this conservation and encode RHIM-containing proteins to prevent RHIM-dependent functions (*Tummers and Green, 2022*; *Riebeling et al., 2022*). Intriguingly, several of these mechanisms are known to be species-specific and differ between humans and mice (*Muscolino et al., 2021*; *Huang et al., 2015*). These data support a model where viruses and host necroptosis are engaged in a molecular arms race that has shaped the function of RIPK3 and RIPK1 across vertebrate species.

Altogether, our combined phylogenetic and functional approaches reveal both conservation and a striking lack of conservation in vertebrate RIPKs and their associated proteins and functions. Such characteristics are common hallmarks of pathways that are required for the innate immune response to pathogens, but are under constant evolutionary pressure to innovate to avoid pathogen antagonism. The strong conservation of NF-κB activation by RIPK1 across species is consistent with the model in which RIPK1 function is central to determining cell fate downstream of TNF and TLR signaling (*Clucas and Meier, 2023*). Moreover, because some pathogens have evolved strategies to inhibit CASP8, RIPK3- and RIPK1-mediated necroptosis is activated as a secondary cell death strategy when CASP8 is inhibited (*Remick et al., 2023*). This places both RIPK3 and RIPK1, critical regulators of necroptosis, at the center of determining cell fate during pathogen infection. The divergence of RIPK3 and RIPK1 across vertebrates identified here, as well as the previous finding that these proteins are evolving under positive selection (*Águeda-Pinto et al., 2021*; *Palmer et al., 2021*; *Cariou et al., 2022*), reveals that pathogens are likely the driving force behind the lineage-specific regulatory mechanisms or functions of these proteins across species that we observe here. Thus, the functional divergence identified in our work highlights the evolutionary innovation that can arise in innate signaling pathways across diverse species and underscores the importance of considering natural diversity when characterizing the innate immune response to pathogens. Our work provides a model for further study on the diversity of innate immune proteins, protein families, and pathways across vertebrate and non-vertebrate species.

## Materials and methods
### Positive selection analysis
Positive selection analysis was performed using three independent methods as we have done previously (*Tsu et al., 2023*; *Stevens et al., 2022*). Nucleotide sequences from the indicated mammalian clade that aligned to full-length human RIPK1–5 were downloaded from NCBI and aligned using ClustalOmega (*Sievers et al., 2011*). Maximum likelihood (ML) tests were performed with codeml in

the PAML software suite (*Yang, 2007*). For PAML, aligned sequences were subjected to ML tests using NS sites models disallowing (M7) or allowing (M8) positive selection. The reported p-values were calculated using a chi-squared test on twice the difference of the log likelihood (lnL) values between the two models using 2 degrees of freedom. We confirmed convergence of lnL values by performing each analysis using two starting omega (dN/dS) values (0.4 and 1.5). Codons evolving under positive selection from PAML analyses have a posterior probability greater than 0.90 using native empirical Bayes (NEB) and Bayes empirical Bayes (BEB) analysis and the F61 codon frequency model. The same nucleotide alignments were used as input for FUBAR (Fast, Unconstrained Bayesian AppRoximation) (*Murrell et al., 2013*) and MEME (Mixed Effects Model of Evolution) (*Murrell et al., 2012*) using the DataMonkey server (*Weaver et al., 2018*). In both cases, default parameters were used, and codons with a signature of positive selection with a p-value of <0.01 are reported. Accession numbers for input sequences, PAML p-values, and sites identified can be found in *Supplementary files 1–3*.

## Gene loss analysis

Gene loss analysis was completed by first building phylogenetic trees for each protein of interest. Phylogenetic trees were generated for RIPK1–5 using previously described methods (*Fu et al., 2012*). Briefly, the kinase domains from human RIPK1 (NP_001341859.1), RIPK2 (NP_003812.1), RIPK3 (NP_006862.2), RIPK4 (NP_065690.2), and RIPK5 (NP_848605.1) were used to query the RefSeq protein database using BLASTp with an e-value cutoff of 2e-26 (*Altschul et al., 1997*). While LRRK1 and LRRK2 share kinase homology with RIPK1–5 and are sometimes known as RIPK6 and -7 (*Lv et al., 2022*), they have a distinct domain structure and function compared to RIPK1–5 and will therefore not be considered as RIP kinases for the purposes of this study. The five resulting sequence lists were consolidated to a single list of unique sequences using Geneious software. These sequences were then aligned using Clustal Omega (*Sievers et al., 2011*) with two refinement iterations implemented in Geneious software. Following alignment, we filtered sequences based on alignment of an N-terminal kinase domain to human RIPK kinase domains and the presence of a C-terminal domain. Proteins with additional N-terminal domains, proteins lacking homology to human kinase domains, and proteins lacking C-terminal domains were removed. Using the aligned and filtered vertebrate RIPK sequences (4550 sequences in total), a tree was built using FastTree (*Price et al., 2009*; *Price et al., 2010*) implemented using Geneious. Vertebrate MAPKK7 sequences were used as an outgroup to help isolate RIP kinases from similar, but distinct, kinases. Using this tree, we were able to identify and extract bona fide RIP kinases. The extracted sequences were again aligned using Clustal Omega and filtered using the criteria above. A second tree was generated using FastTree. From this tree, we extracted individual sequence lists for RIPK1–5 to determine which vertebrate species did or did not have each RIPK. This method was repeated for MLKL, TRIF, FADD, CASP8 and 10, NOD 1 and 2, and IRF6. The list of species and lists of accession numbers for each protein can be found in *Supplementary files 4 and 5*. A list of outgroups used for each analyzed protein can be found in *Supplementary file 6*.

## Plasmids and constructs

Coding sequences for human RIPK1 (Addgene #78834), human RIPK2 (ORFeome ID #4886), human RIPK3 (Addgene #78804), mouse RIPK1 (Addgene #115341), and mouse RIPK3 (Addgene #78805) were cloned into apcDNA5/FRT/TO backbone (Invitrogen, Carlsbad, CA, USA) with an N-terminal V5 tag and linker using Gibson Assembly (New England Biolabs, Ipswich, MA, USA). Human RIPK4 (NP_065690.2) and RIPK5 (NP_848605.1), and nonhuman RIPK1 C-termini (cat, XP_023109490.2; pig, XP_003128209.1; chicken, NP_989733.3; lizard, XP_003224434.1; lamprey, XP_032813622.1; zebra mussel, XP_052232845.1) and RIPK3 (cat, XP_003987615.3; pig, XP_001927459.3; lizard, XP_003223896.2; turtle, XP_037771912.1; lamprey, XP_032816533.1) homologues were ordered from Twist Biosciences (San Francisco, CA, USA) and were cloned along with RHIM mutants targeting the core tetrad into the pcDNA5/FRT/TO backbone (Invitrogen, Carlsbad, CA, USA) with an N-terminal V5 tag and linker using Gibson Assembly (New England Biolabs, Ipswich, MA, USA). Kinase mutants for human proteins RIPK1 (D138N), RIPK2 (D146N), RIPK3 (D142N), RIPK4 (D143N), and RIPK5 (D145N), and RHIM mutants for RIPK1 (IQIG542AAAA) and RIPK3 (VQVG461AAAA) were generated using Gibson Assembly. A list of primers used is found in *Supplementary file 9*.

## Cell culture and transient transfections

HEK293T cells (obtained from ATCC) and the generated HEK293T RIPK1 KO cells were maintained at a low passage number to maintain less than 1 year since purchase, acquisition, or generation. Both cell lines were grown in complete media containing DMEM (Gibco, Carlsbad, CA, USA), 10% FBS (Peak Serum, Wellington, CO, USA), and 1% penicillin/streptomycin (Gibco, Carlsbad, CA, USA). A day prior to transfection, HEK293T and RIPK1 KO cells were seeded into 24-well plates with 500 μL complete media or 96-well plates with 80 μL complete media. Cells were transiently transfected with 500 ng total DNA and 1.5 μL of TransIT-X2 (Mirus Bio, Madison, WI, USA) in 100 μL Opti-MEM (Gibco, Carlsbad, CA, USA) for a 24-well plate or 100 ng DNA and 0.3 μL TransIT-X2 in 10 μL Opti-MEM for a 96-well plate. DNA and TransIT-X2 were incubated at room temperature for 25–30 min, then added dropwise to the appropriate well. Cells were harvested or analyzed at 18–22 hr post-transfection. Cell lines are routinely confirmed to be mycoplasma-free by PCR analysis (using ATCC product #30-1012K). HEK293T cells were authenticated by STR analysis (SBP Genomics Core, La Jolla, CA, USA).

## Generation of knockout cell lines

RIPK1 knockout HEK293T cells were generated using CRISPR/Cas9 as previously described (*Stevens et al., 2022*; *Tsu et al., 2021*). Briefly, plasmids were generated in order to produce lentivirus-like particles containing the CRISPR/Cas9 machinery and guide RNA targeting exon 5 (ENSE00003586162) of RIPK1. The protocol for the molecular cloning of this plasmid was adapted from Feng Zhang (*Sanjana et al., 2014*) using the transfer plasmid pLB-Cas9 (gift from Feng Zhang, Addgene plasmid # 52962). We designed the gRNA target sequence using the web tool CHOPCHOP (*Labun et al., 2019*), available at https://chopchop.cbu.uib.no/, and synthesized oligonucleotides from Integrated DNA Technologies (San Diego, CA, USA). The synthesized oligonucleotide pair was phosphorylated and annealed using T4 Polynucleotide Kinase (NEB M0201S) with T4 Ligation Buffer (NEB). Duplexed oligonucleotides were ligated into dephosphorylated and BsmBI-digested pLB-Cas9 using the Quick Ligase kit (NEB M2200S) to generate transfer plasmid with RIPK1 guide sequence. To generate lentivirus-like particles, this transfer plasmid was transfected alongside two packaging plasmids, pMD2.G (gift from Didier Trono, Addgene plasmid # 12259) and psPAX2 (gift from Didier Trono, Addgene plasmid # 12260), into HEK293T cells with a 1:1:1 stoichiometry. Forty-eight hours post-transfection, supernatant was harvested and syringe-filtered (0.45 μm). Supernatant containing sgRNA-encoding lentivirus-like particles was then used to transduce HEK293T cells. Transduced cells were cultured in growth media for 48 hr, then cultured in growth media supplemented with 1 μg/mL puromycin for 72 hr. Using limiting dilution in 96-well plates, the monoclonal cell line was then obtained. RIPK1 knockout was confirmed by Sanger sequencing and by western blot using an α-RIPK1 antibody.

## NF-κB and IFN luciferase reporter assays

To quantify NF-κB and IFN activation, we used the Dual-Glo Luciferase Assay System E2920 (Promega, USA). WT or RIPK1 KO HEK293T cells were seeded in a white 96-well plate and transfected with firefly luciferase fused to either the NF-κB response element (pGL4.32, Promega) or the human IFN-beta promoter (IFN-Beta-pGL3, Addgene #102597), renilla luciferase fused to herpes simplex virus thymidine kinase promoter (pTK-Renilla, Thermo Scientific), and the indicated RIPK. Firefly luciferase was used as the primary reporter with renilla luciferase being a normalization control. Eighteen to 24 hr after transfection, Dual-Glo Luciferase Assay Reagent was added to each transfected well as well as three to six untransfected wells to serve as a negative control. Following a 10 min incubation, the firefly luciferase signal was measured using a BioTek Cytation imaging reader with Gen5 software (Agilent Technologies, San Diego, CA, USA). Then, Dual-Glo Stop & Glo reagent was added to each well. Following a 10 min incubation, the renilla luciferase signal was measured. The background luminescence signal from the buffer-treated untransfected conditions was subtracted from other conditions, firefly values were normalized to renilla values, and all samples were normalized to the empty vector control condition. Values are reported as NF-κB activity (firefly/renilla).

## Cell death assay

Cell viability was measured using the ReadyProbes Cell Viability Imaging Kit, Blue/Green (Thermo Fisher Scientific). HEK293T cells were seeded in 80 μL of DMEM+/+ (see 'Cell culture and transient transfections') in a clear 96-well plate and transfected with the indicated plasmids or a vector

control. Eighteen hours post-transfection, cells were stained using NucBlue Live reagent, which stains all nuclei, and NucGreen Dead reagent, which stains only dead cells. A 2× concentrated mix (4 drops/mL of each dye) was made and added to wells to 1× concentration. Cells were incubated for 20 min at room temperature. Fluorescence for NucBlue (DAPI, excitation 377/20, emission 447/20) and NucGreen (GFP, excitation 469/20, emission 525/20) was read using a BioTek Cytation imaging reader with Gen5 software (Agilent Technologies, San Diego, CA, USA). Unstained wells were read and used as controls for background fluorescence. GFP values were normalized to DAPI values, and all samples were normalized to the vector control condition. Values are reported as cytotoxicity (GFP/DAPI).

## Immunoblotting and antibodies

Eighteen to 22 hr post-transfection, cells were washed with 1× PBS and lysed with boiling 1× NuPAGE LDS sample buffer containing 5% β-mercaptoethanol at room temperature for 5 min and then at 98°C for 7–10 min. Lysates were separated by SDS-PAGE (4–15% Bis-Tris gel; Life Technologies, San Diego, CA, USA) with 1× MOPS buffer (Life Technologies, San Diego, CA, USA). Proteins were transferred onto a nitrocellulose membrane (Life Technologies, San Diego, CA, USA) and blocked with PBS-T containing 5% bovine serum albumin (BSA) (Spectrum, New Brunswick, NJ, USA). Membranes were incubated with the indicated rabbit primary antibodies diluted with 5% BSA and PBS-T at 1:1000 overnight at 4°C (α-V5 clone D3H8Q, α-GAPDH clone 14C10, α-RIPK1 clone E8S7U XP, α-RIPK2 clone D10B11, α-RIPK3 clone E1Z1D, α-RIPK4 #12636; Cell Signaling Technology, Danvers, MA, USA). Membranes were rinsed three times with PBS-T, incubated with HRP-conjugated rabbit secondary antibody diluted at 1:10,000 with 5% BSA and PBS-T for 30 min at room temperature, and developed with SuperSignal West Pico PLUS Chemiluminescent Substrate (Thermo Fisher Scientific, Carlsbad, CA, USA).

## Statistical analysis

Statistical analyses were completed using GraphPad Prism 10 software. Tests were performed as indicated. Error bars were calculated using SEM.

## Acknowledgements

This work was supported by National Institutes of Health grant R35 GM133633 (to MDD); Pew Biomedical Scholars Program (to MDD); Burroughs Wellcome Investigators in the Pathogenesis of Infectious Disease Program (to MDD); and BrightSpinnaker Fellowship (to CMR). We thank all members of the Daugherty lab for helpful discussions and Scott Biering, Ryan Langlois, and Patrick Mitchell for their critical reading of this manuscript.

## Additional information

### Funding

| Funder | Grant reference number | Author |
|---|---|---|
| National Institute of General Medical Sciences | R35 GM133633 | Matthew D Daugherty |
| Pew Charitable Trusts | Biomedical Scholars Program | Matthew D Daugherty |
| Burroughs Wellcome Fund | Investigators in the Pathogenesis of Infectious Disease program | Matthew D Daugherty |
| BrightSpinnaker Fellowship | | Charles M Rezanka |

The funders had no role in study design, data collection and interpretation, or the decision to submit the work for publication.

## Author contributions
Elizabeth J Fay, Conceptualization, Formal analysis, Supervision, Investigation, Visualization, Writing – original draft, Writing – review and editing; Kolya Isterabadi, Charles M Rezanka, Jessica Le, Investigation, Writing – review and editing; Matthew D Daugherty, Conceptualization, Supervision, Funding acquisition, Writing – original draft, Writing – review and editing

## Author ORCIDs
Elizabeth J Fay (iD) https://orcid.org/0000-0003-1712-757X
Matthew D Daugherty (iD) https://orcid.org/0000-0002-4879-9603

Reviewer #2 (Public review): https://doi.org/10.7554/eLife.102301.3.sa1
Reviewer #3 (Public review): https://doi.org/10.7554/eLife.102301.3.sa2
Author response https://doi.org/10.7554/eLife.102301.3.sa3

# Additional files

## Supplementary files
Supplementary file 1. Accession numbers for genes used in PAML analysis.

Supplementary file 2. p-Values calculated by PAML.

Supplementary file 3. RIPK3 sites identified by PAML, FUBAR, and MEME as evolving under positive selection.

Supplementary file 4. Accession numbers for proteins analyzed in *Figure 1C*.

Supplementary file 5. List of species included in analysis in *Figure 1C*.

Supplementary file 6. List of proteins used as outgroups for analysis in *Figure 1C*.

Supplementary file 7. Expression of CASP8 and ZBP1 in reptile species.

Supplementary file 8. List of RHIM variants identified in vertebrate RIPK3 proteins.

Supplementary file 9. Primers used to generate plasmid DNA constructs used in this study.

MDAR checklist

## Data availability
All data generated or analyzed during this study are included in the manuscript and supporting files. Values used to generate plots are included as source data for each figure. Western blot images are provided as raw, unedited, uncropped images as source data.

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
