## [Editor Report · eLife Assessment]

This **important** study provides **compelling** evidence for the evolutionary diversification and conserved NFκB-inducing function of RHIM-containing RIP kinase proteins across animal lineages, combining thorough bioinformatic analysis with functional assays in human cells. The findings are of broad interest to immunologists and evolutionary biologists, though some novel observations would benefit from deeper conceptual integration.

---

## [Referee Report · Reviewer #2 (Public review)]

Summary:

By combining bioinformatical and experimental approaches, the authors address the question why several vertebrate lineages lack specific genes of the necroptosis pathway, or those that regulate the interplay between apoptosis and necroptosis. The lack of such genes was already known from previous publications, but the current manuscript provides a more in-depth analysis and also uses experiments in human cells to address the question of functionality of the remaining genes and pathways. A particular focus is placed on RIPK3/RIPK1 and their dual roles in inducing NFkB and/or necroptosis.

Strengths:

The well documented bioinformatical analyses provide a comprehensive data basis of the presence/absence of RIP-kinases, other RHIM proteins, apoptosis signaling proteins (FADD,CASP8,CASP10) and some other genes involved in these pathway. Several of these genes are known to be missing in certain animal lineages, which raises the question why their canonical binding partners are present in these species. By expressing several such proteins (both wildtype and mutants destroying particular interaction regions) in human cells, the authors succeed in establishing a general role of RIPK3 and RIPK1 in NFkB activation. This function appears to be better conserved and more universal than the necroptotic function of the RHIM proteins. The authors also scrutinize the importance of the kinase function and RHIM integrity for these separate functionalities.

Weaknesses:

A weakness of the presented study is the experimental restriction to human HEK293 cells. There are several situations where the functionality of proteins from distant organisms (like lampreys or even mussels) in human cells is not necessarily indicative of their function in native context. In some cases, these problems are addressed by co-expressing potential interaction partners, but not all of these experiments are really informative. However, I agree with the authors that it is not possible to perform all the experiments in native cells, and that comparing all proteins in the same (human) cell type allows for a better comparison.

The conclusions drawn by the authors are supported by convincing evidence. I have no doubts that this study will be very useful for future studies addressing the evolution of necroptosis and its regulation by NFkB and apoptosis.

---

## [Referee Report · Reviewer #3 (Public review)]

In this study, the authors employ both computational and experimental methods to reveal functional conservation of RIP family kinases and associated proteins in animals, with particular focus on mammals and other major groups of vertebrates. The bionformatic part of the work involves genomic data from diverse animal groups, providing insightful data on loss and duplications patterns for RIP and other necroptosis-related genes, and positive selection signals for RIPK1/3 genes in certain mammalian clades. These findings are then extensively used for selecting species and RHIM tetrad candidates for further experiments, in which the authors demonstrate different modes of functional conservation for RIPK proteins in necroptosis and NF-kB signaling across vertebrate species.

As an only major drawback, I would mention several important findings which the authors make in the course of their research but do not pursue further in the experimental part of the paper. These include:

• An additional copy for RIPK2 (RIPK2B) found in monotremes and non-mammalian vertebrates and its functions;

• The entire diversity of RHIM functional tetrad variants; of particular interest here are IQFG and IQLG tetrads specific for bats, which are known to harbor human-affecting viruses and were demonstrated to have their RIPK1/3 genes under positive selection in this study;

• Functions and involvement of RIPK3 protein in NF-kB pathway in lampreys;

• The mode of NF-kB activation in non-mammalian species retaining ZBP1 copies.

Further elucidation of some or all of these points in the experimental part would facilitate conceptualizing the paper's numerous findings, which otherwise might appear insufficiently scrutinized. On the other hand, I agree that at least some of them require separate studies to be elucidated in. Given the importance of the results presented in this paper, I believe these points will be further addressed in future works.

---

## [Author Response]

The following is the authors’ response to the original reviews

**Public Reviews:**

**Reviewer #1 (Public review):**
The manuscript titled "Evolutionary and Functional Analyses Reveal a Role for the RHIM in Tuning RIPK3 Activity Across Vertebrates" by Fay et al. explores the function of RIPK gene family members across a wide range of vertebrate and invertebrate species through a combination of phylogenomics and functional studies. By overexpressing these genes in human cell lines, the authors examine their capacity to activate NF-κB and induce cell death. The methods employed are appropriate, with a thorough analysis of gene loss, positive selection, and functionality. While the study is well-executed and comprehensive, its broader relevance remains limited, appealing mainly to specialists in this specific field of research. It misses the opportunity to extract broader insights that could extend the understanding of these genes beyond evolutionary conservation, particularly by employing evolutionary approaches to explore more generalizable functions.Major comments:The main issue I encounter is distinguishing between what is novel in this study and what has been previously demonstrated. What new insights have been gained here that are of broader relevance? The discussion, which would be a good place to do so, is very speculative and has little to do with the actual results. Throughout the manuscript, there is little explanation of the study's importance beyond the fact that it was possible to conduct it. Is the evolutionary analysis being used to advance our understanding of gene function, or is the focus merely on how these genes behave across different species? The former would be exciting, while the latter feels less impactful.

We thank the reviewer for the positive feedback. With regard to the major comment, we have now made changes throughout the revised manuscript to highlight the novel insights that emerge from our work, as well as the importance of using evolutionary and functional analyses to understand gene function.

**Reviewer #2 (Public review):**
Summary:By combining bioinformatical and experimental approaches, the authors address the question of why several vertebrate lineages lack specific genes of the necroptosis pathway or those that regulate the interplay between apoptosis and necroptosis. The lack of such genes was already known from previous publications, but the current manuscript provides a more in-depth analysis and also uses experiments in human cells to address the question of the functionality of the remaining genes and pathways. A particular focus is placed on RIPK3/RIPK1 and their dual roles in inducing NFkB and/or necroptosis.Strengths:The well-documented bioinformatical analyses provide a comprehensive data basis of the presence/absence of RIP-kinases, other RHIM proteins, apoptosis signaling proteins (FADD, CASP8, CASP10), and some other genes involved in these pathways. Several of these genes are known to be missing in certain animal lineages, which raises the question of why their canonical binding partners are present in these species. By expressing several such proteins (both wildtype and mutants destroying particular interaction regions) in human cells, the authors succeed in establishing a general role of RIPK3 and RIPK1 in NFkB activation. This function appears to be better conserved and more universal than the necroptotic function of the RHIM proteins. The authors also scrutinize the importance of the kinase function and RHIM integrity for these separate functionalities.Weaknesses:A major weakness of the presented study is the experimental restriction to human HEK293 cells. There are several situations where the functionality of proteins from distant organisms (like lampreys or even mussels) in human cells is not necessarily indicative of their function in the native context. In some cases, these problems are addressed by co-expressing potential interaction partners, but not all of these experiments are really informative.A second weakness is that the manuscript addresses some interesting effects only superficially. By using host cells that are deleted for certain signaling components, a more focussed hypothesis could have been tested.

Thus, while the aim of the study is mostly met, it could have been a bit more ambitious. The limited conclusions drawn by the authors are supported by convincing evidence. I have no doubts that this study will be very useful for future studies addressing the evolution of necroptosis and its regulation by NFkB and apoptosis.

We thank the reviewer for the positive feedback. We agree that our study is limited by using HEK293 cells. However, we do not have appropriate cell lines for all species analyzed and therefore wished to use a single system to test all effects. As the reviewer points out, we do co-express when possible, and are careful in the manuscript to not overextend our conclusions. We, like the reviewer, believe that many of the intriguingly findings in this study, which was intended to cover a broad range of species, will be useful for more in-depth studies in a given species.

**Reviewer #3 (Public review):**
This important study provides insights into the functional diversification of RIP family kinase proteins in vertebrate animals. The provided results, which combine bioinformatic and experimental analyses, will be of interest to specialists in both immunology and evolutionary biology. However, the computational part of the methodology is insufficiently covered in the paper and the experimental results would benefit from including data for additional species.

We thank the reviewer for the positive feedback. As described below, we have now addressed the concerns about the description of the computational methods.

(1) In the Methods section concerning gene loss analysis, the authors refer to the 'Phylogenetic analysis' section for details of RIPK sequence acquisition and alignment procedure. This section is missing from the manuscript as provided. In its absence, it is hard for the reviewer to provide relevant comments on gene presence/absence analysis.

We have expanded the gene loss analysis methods to be more comprehensive.

(2) In the same section, the authors state that gene sequences were filtered and grouped based on the initial gene tree pattern (lines 448-449). How exactly did the authors filter the non-RIP kinases and other irrelevant homologs from the gene trees? Did they consider the reciprocal best (BLAST) hit approach or similar approaches for orthology inference? Did they also encounter potential pseudogenes of genes marked as missing in Figure 1C? Will the gene trees mentioned be available as supplementary files?

We have expanded the gene loss analysis methods to be more comprehensive.

(3) The authors state the presence of additional RIPK2 paralog in non-therian vertebrates.The ramifications of this paralog loss in therians are not discussed in the text, although RIPK2 is also involved in NF-kB activation. In addition, the RIPK2B gene loss pattern is shunned from Figure 1C to Supplementary Figure 4, despite posing comparable interest to the reader.

We are also intrigued by the RIPK2/RIPK2B data and felt it important to include our findings here, however we do not have functional data for RIPK2B at this point and feel it is better suited for a separate study. We therefore focused both the title and the main figures on RIPK3, for which we have functional data.

(4) The authors present evidence for (repeated) positive selection in both RIPK1 and RIPK3 in bats; however, neither bat RIPK1/3 orthologs nor bat-specific RHIM tetrad variants (IQFG, IQLG) are considered in the experimental part of the work.

We included a tetrad variant (VQFG) that is found in bats and multiple other species. We wanted to test a wide range of variant amino acids, so testing both IQFG (found only in bats) and VQFG (found in bats and multiple other diverse species) was not of high importance.

(5) The authors present gene presence/absence patterns for zebra mussels as an outgroup of vertebrate species analyzed. From the evolutionary perspective, adding results for a closer invertebrate group, such as lancelets, tunicates, or echinoderms, would be beneficial for reconstructing the evolutionary progression of RIPK-mediated immune functions in animals.

In our initial analyses, we searched for RIPK-like proteins in cnidarians, arthropods, nematodes, amoeba, and spiralia, with only spiralia species containing proteins with substantial homology to vertebrate RIPK1 proteins, as defined by a homologous N-terminal kinase domain and C-terminal RHIM and death domain. We have expanded this analysis to include lancelets, tunicates, and echinoderms and found several lancelet species with RIPK1 like proteins. These data have been added to the manuscript.

(6) In the broader sense, the list of non-mammalian species included in the study is not explained or substantiated in the text. What was the rationale behind selecting lizards, turtles, and lampreys for experimental assays? Why was turtle RIPK3 but not turtle RIPK1CT protein used for functional tests? Which results do the authors expect to observe if amphibian or teleost RIPK1/3 are included in the analysis, especially those with divergent tetrad variants?

We have added additional text to define our rationale for selecting which species were tested.

(7) For lamprey RIPK3, the observed NF-kB activity levels still remain lower than those of mammalian and reptilian orthologs even after catalytic tetrad modification. In the same way, switching human RIPK3 catalytic tetrad to that of lamprey does not result in NF-kB activation. What are the potential reasons for the observed difference? Does it mean that lamprey's RIPK3 functions in NF-kB activation are, at least partially, delegated to RIPK1?

The function of lamprey RIPK3 is intriguing, albeit unknown. The reduced activation in human cells may be due to an incompatibility between lamprey RIPK3 and human NF-kB machinery, or it may not function in NF-kB at all. Considering that lamprey do not have other components of the known mammalian necroptosis pathway, it is unclear what function RIPK3 would serve in these species. It is possible lamprey may have a necroptosis pathway that is RIPK3-dependent but distinct from the mammalian pathway. It is an interesting question for future study.

(8) In lines 386-388, the authors state that 'only non-mammalian RIPK1CT proteins required the RHIM for maximal NF-kB activation', which is corroborated by results in Figure 4B. The authors further associate this finding with a lack of ZBP1 in the respective species (lines 388-389). However, non-squamate reptiles seem to retain ZBP1, as suggested by

Supplementary Table 1. Given that, do the authors expect to observe RHIM-independent (maximal) NF-kB activation in turtles and crocodilians or respective RIPK1CT-transfected cells?

While turtles and crocodiles do retain ZBP1, it is still unclear if they are able to activate ZBP1/RIPK3/MLKL-dependent necroptosis similar to mammals, especially given the divergence in the turtle ZBP1 RHIMs seen in Figure 4C. Future studies will be needed to further test our hypotheses and to continue to characterize innate immune function and evolution across a range of vertebrate species.

**Recommendations for the authors:**

**Reviewer #1 (Recommendations for the authors):**
Minor comments:(1) The title is somewhat restrictive, as it only mentions RIPK3, despite the manuscript covering a broader range of RIPKs and associated proteins.

We agree that a title that encompasses both the breadth of our study and the depth with which we analyzed RIPK3 would be ideal. However, we were unable to come up with a succinct title that conveyed both points appropriately, so opted for one that focused on our RIPK3 insights.

(2) Several supplementary figures contain valuable information that could be incorporated into the main figures for greater clarity and emphasis.

We agree that many interesting pieces of data are in the supplement. We felt it was important to include those data in the manuscript, but also wanted to keep the main manuscript figures as focused as possible.

**Reviewer #2 (Recommendations for the authors):**
(1) I do not fully agree with the claim that caspase-8 is absent from fish. I briefly repeated this part of the analysis and found several fish proteins that cluster with caspase-8 rather than caspase-10 or cFLIP. From the method section, it does not really become clear how the Casp8/Casp10/cFLIP decision was made, and particularly, how cases were addressed where Genew predate the caspase-8/caspase-10 split. To name just a few examples, the authors might check uniprot:A0A444UA91, W5MXS4, or A0A8X8BKJ8 for being fish Caspase-8 candidates.

We thank the reviewer for their critical analysis. CASP8 and CASP10 are very similar proteins in humans. We are distinguishing between the two based on vertebrate phylogeny with outgroup proteins (CASP2, CASP9, and CFLAR, see tree in Author response image 1 below) to help define the CASP8/CASP10 clade. Once we isolate CASP8/10, we build an additional tree to distinguish CASP8 and CASP10. Using this method, all fish CASP8/10-like proteins cluster with the mammalian CASP10 clade rather than the CASP8 clade, despite many fish proteins being annotated as CASP8 or CASP8-like. We do acknowledge that, because of the similarities between CASP8 and CASP10, there are likely proteins that can fall in either clade depending on which outgroups are included. To this end, we have updated our gene loss figure to only denote whether a species has no CASP8/10, a single CASP8/10 protein, or both CASP8 and CASP10. We have also updated our methods to better define how we completed our analyses.

(2) While analyzing which RIPK3 protein causes cell death (lines 188ff), the underlying assumption is that the heterologous RIPK3 proteins can interact with human MLKL and activate it by phosphorylation. No attempts are being made to check if MLKL actually gets phosphorylated, and this issue is also not discussed. In Figure 2C, cell death is either measured by RIPK3 overexpression alone or by the additional overexpression of ZBP1 and MLKL. However, it is not shown if in all cases all the transfected proteins are expressed at a comparable level, or if the observed cell death might be caused by MLKL/ZBP1 overexpression alone.

Cell death is dependent on expression of ZBP1, MLKL, and RIPK3, as shown in

Supplementary Figure 6. We have attempted to detect phospho-MLKL via western blot. However, in these overexpression assays, we are able to detect phospho-MLKL in the presence of RIPK3 and MLKL alone, independent of activation of cell death. In fact, we see reduced phospho-MLKL and reduced expression of MLKL overall when ZBP1, MLKL, and RIPK3 are added, presumably due to cell death induced in these conditions (see blot in Author response image 2 below). We therefore felt these data were of limited use here.

**Author response image 2. sa3fig2:** 

(3) The manuscript describes a well-documented bioinformatical analysis and acknowledges the body of earlier published work on necroptosis evolution and associated gene losses. However, when discussing the RHIM-related aspects, the authors do not mention previous publications on RHIM conservation in invertebrates and even fungal proteins such as Het-S. They also fail to mention/discuss the amyloid-forming properties of RHIMs, which I consider crucial for understanding the function of RHIM-containing proteins.

We thank the reviewer for their insight. We have added additional points on both RHIM conservation and amyloid formation.

(4) Related to the above issue: In lines 226ff, the induction of NFkB by RIPK3 overexpression is described. While RIPK3 from other mammals requires endogenous (human) RIPK1 to be present, lizard and turtle RIPK3 do not require human RIPK1 but *do* require functional RHIMs. It is not checked (or at least discussed) if RHIM amyloid formation is required, nor if the RHIM of the heterologous RIPK3 might act through interaction with endogenous (human) RIPK3.

We and others (PMID: 29073079) did not detect RIPK3 protein in HEK293T cells. This, combined with the requirement for exogenous RIPK3 to activate cell death, indicate that endogenous RIPK3 is not contributing to these assays.

(5) In lines 275ff, the authors observe that RIPK1s from other mammalian species do not require the RHIM for NFkB activation, while RIPK1 from non-mammalian species do require the RHIM. I wonder why the (in my opinion) most obvious explanation is not addressed: Maybe the mammalian RIPK1 proteins are similar enough to the human one so that they can signal on their own, while the more distant RIPK1 cannot and thus require human RIPK1 (associated via RHIMs) for NFkB activation? Since the authors used RIPK1-deficient cells in previous experiments, wouldn't it make sense to test them here, too?

It is intriguing that the more diverged RIPK1 species require the RHIM for NF-kB signaling. In Supplementary Figure 12, we do test the mammalian and non-mammalian proteins in RIPK1 KO cells and all proteins are able to activate NF-kB. So while nonmammalian RIPK1 signaling is dependent on the RHIM, it is independent of endogenous RIPK1.

Minor comments:(1) In the legend of Figure 1, there is a typo "heat amp".

This typo has now been corrected.

(2) In Figure 3A, the term "FUBAR" is not explained at all.

FUBAR has now been defined in the methods section.

**Reviewer #3 (Recommendations for the authors):**
A few typos and graph inconsistencies have been encountered in the course of the manuscript, e.g.:(1) Line 168: 'heat amp' -> 'heat map'.(2) Lines 290-291: 'known mediate' -> 'known to mediate' (?)

We thank the reviewer for catching these mistakes. They have been corrected.

(3) Supplementary Figure 12: Are human RIPK1 results presented in both 'mammalian' and 'non-mammalian' parts of the figure? If so, why do human data differ between the graphs?

Mammalian and non-mammalian data were collected in separate experiments with human RIPK1 used as a control for both. The human data shown in the two graphs represent two separate experiments.